# Is carotid artery atherosclerosis associated with poor cognitive function assessed using the Mini-Mental State Examination? A systematic review and meta-analysis

Rayan Anbar [1,2] Salahaden R Sultan [1] Lamia Al Saikhan [3]
Mohammed Alkharaiji [4] Nishi Chaturvedi,[2] Rebecca Hardy,[5] Marcus Richards,[2]
Alun Hughes [2,6]

For numbered affiliations see end of article.

**Correspondence to**
Professor Alun Hughes;
alun.hughes@ucl.ac.uk

## ABSTRACT

**Objectives** To determine associations between carotid atherosclerosis assessed by ultrasound and the Mini-Mental State Examination (MMSE), a measure of global cognitive function.

**Design** Systematic review and meta-analysis.

**Methods** MEDLINE and EMBASE databases were searched up to 1 May 2020 to identify studies assessed the associations between asymptomatic carotid atherosclerosis and the MMSE. Studies reporting OR for associations between carotid plaque or intima-media thickness (cIMT) and dichotomised MMSE were meta-analysed. Publication bias of included studies was assessed.

**Results** A total of 31 of 378 reviewed articles met the inclusion criteria; together they included 27 738 participants (age 35–95 years). Fifteen studies reported some evidence of a positive association between measures of atherosclerosis and poorer cognitive performance in either cross-sectional or longitudinal studies. The remaining 16 studies found no evidence of an association. Seven cross-sectional studies provided data suitable for meta-analysis. Meta-analysis of three studies that assessed carotid plaque (n=3549) showed an association between the presence of plaque and impaired MMSE with pooled estimate for the OR (95% CI) being 2.72 (0.85 to 4.59). An association between cIMT and impaired MMSE was reported in six studies (n=4443) with a pooled estimate for the OR (95% CI) being 1.13 (1.04 to 1.22). Heterogeneity across studies was moderate to small (carotid plaque with MMSE, $I^2$=40.9%; cIMT with MMSE, $I^2$=4.9%). There was evidence of publication bias for carotid plaque studies (p=0.02), but not cIMT studies (p=0.2).

**Conclusions** There is some, limited cross-sectional evidence indicating an association between cIMT and poorer global cognitive function assessed with MMSE. Estimates of the association between plaques and poor cognition are too imprecise to draw firm conclusions and evidence from studies of longitudinal associations between carotid atherosclerosis and MMSE is limited.

**PROSPERO registration number** CRD42021240077.

## STRENGTHS AND LIMITATIONS OF THIS STUDY

⇒ We performed a systematic review on 31 published studies assessed the association between carotid atherosclerosis and global cognition.

⇒ The association between carotid atherosclerosis and global cognition was meta-analysed on 7992 asymptomatic patients.

⇒ There was some evidence from cross-sectional studies that carotid plaque and carotid intima–media thickness were associated with impaired Mini-Mental State Examination (MMSE).

⇒ Methods used and characteristics of participants across studies were different which makes it difficult to compare all studies.

⇒ This review only included studies which used MMSE as a cognitive function test which may result in bias or an underestimation of decline in cognitive state.

## INTRODUCTION

Improvements in life expectancy have led to a dramatic increase in the worldwide burden of age-related diseases, notably atherosclerotic cardiovascular disease, and cognitive decline and dementia.[1]

Atherosclerosis is a chronic inflammatory disease, which begins in childhood.[2] With time and progression of disease, fibrofatty plaques develop which may result in alterations in the elastic properties of the artery, narrowing of the lumen and resultant effects on haemodynamics, or thromboembolic consequences. Atherosclerosis affects most large arteries, including the common carotid artery (CCA) and its major branch, the internal carotid artery.[3] Carotid artery atherosclerosis is common[4] and may be an important cause of cerebral ischaemia and cognitive impairment. In carotid stenosis, embolisation and hypoperfusion are thought to be the major

processes that cause cognitive impairment,[5] but alterations in cerebral haemodynamics secondary to atherosclerosis could also make a contribution that is potentially reversible.[6 7]The assessment of carotid disease can be performed using a range of imaging methods, (eg, MRI, CT, angiography and ultrasound),[8] ultrasound is the most commonly used,[9–12] as it is inexpensive, non-invasive and widely available.

Cognitive function refers to internal mental processes which help people to think, make decisions and solve problems,[13] and a decline in cognitive function can affect patients' health, daily routine, learning new things, speech and writing abilities, and their ability to live independently. A decline in cognitive function is common with advancing age,[14] and severe cognitive impairment and dementia are placing an increasingly heavy social, emotional and economic burden on society.[15] Dementia is a syndrome that results from diseases of the brain which are usually chronic or progressive.[15] Dementia is characterised by severe deficits in cognition that have a substantial effect on the activities of daily life. These impairments in cognition are also often accompanied by changes in emotional control, social behaviour and motivation.[15] Mild cognitive impairment (MCI) is defined as an impairment of cognition that is greater than expected based on an individual's age and education which does not appreciably affect daily functioning. Nevertheless, despite its denomination as mild, it is well established that people with MCI are at a high risk of developing dementia, particularly of the Alzheimer type[16]A comprehensive assessment of neuropsychological cognitive performance at the domain level is time consuming and most studies use brief cognitive screening tools.[17]

The most widely used tool for cognitive testing is the Mini-Mental State Examination (MMSE).[18–20] MMSE contains 11 tasks and covers 7 domains: visuospatial skills, language, concentration, working memory, memory recall and orientation and is scored out of a maximum 30 points.[21] The most common cut-points to detect dementia with the MMSE are ≤23 or ≤24, although higher and lower cut-points have been used in some studies.[19 20]

Several studies have used a combination of carotid ultrasound and MMSE to examine the associations between carotid atherosclerosis and cognitive performance, either cross-sectionally or longitudinally; however, these studies have yielded conflicting results. For example, the Rotterdam Study[22] reported that CCA intima–media thickness (cIMT) was associated with increased risk of Alzheimer's disease, but not vascular dementia, whereas carotid plaques were not associated with either Alzheimer's disease or vascular dementia. In contrast, the Framingham Offspring Cohort study[23] reported that cIMT was not associated with any measure of cognition, although there was evidence of an association between cIMT and impaired verbal memory and nonverbal memory, and that carotid stenosis ≥50% was associated with impaired executive function, but not verbal and non-verbal memory.

The aim of this study, therefore, was to systematically review evidence of an association between carotid atherosclerosis assessed by ultrasound and global cognitive function assessed using the MMSE, also to perform a meta-analysis of quantitative measures of association between carotid atherosclerosis and global cognitive function.

## METHODS

This study was conducted according to the 'Preferred Reporting Items for Systematic Reviews and Meta-Analyses' (PRISMA) statement.[24] Patients or the public were not involved in the design, or conduct, or reporting, or dissemination plans of our research.

### Search strategy

The search was conducted systematically (up to 1 May 2020) by two authors independently in two online databases: MEDLINE and EMBASE Classic+EMBASE following training and support from a information specialist. A combination of synonyms and related words and text word searching was used to comprehensively extract all relevant articles. A combination of indexed (MesH) terms and keyword searches in titles and abstracts was used - (1) cognition, cognitive function, dementia, Alzheimer disease; (2) atherosclerosis, IMT, plaque; (3) carotid arteries and (4) all searches were limited to (English language and humans). The results from databases were exported to an Endnote library, and any duplicates results were identified and removed before screening the records. The records of non-relevant articles were excluded by screening titles and abstracts by three authors working independently. The remaining publications were assessed by screening the full texts for eligibility and were retrieved and double screened. Discrepancies were reviewed and resolved through consensus. Full search strategies for all databases are available in online supplemental files 1 and 2.

### Inclusion criteria

Relevant studies were required to include the following: (1) Human adults (ie, a person older than 19 years of age)[25] (2) cIMT and/or carotid artery plaque measured using ultrasound and (3) MMSE as a global cognitive function test.

### Exclusion criteria

The exclusion criteria as follows: (1) Adults following stroke (if results in stroke patients were reported separately these were excluded, or if the study only included participants with stroke the entire study was excluded); (2) Patients who had undergone carotid surgical and non-surgical intervention; (3) Studies that used medical imaging modalities other than ultrasound to assess the presence of plaque and to measure carotid atherosclerosis; (4) Review articles, conference abstracts, case reports, letters to the editor or commentary articles.

## Data extraction

Data extraction from the included studies were performed by three researchers independently using standardised forms; any discrepancies were resolved by consensus. The extracted data from the eligible studies consisted of study characteristics such as author, year of publication, sample size, characteristics of participants (age and gender), study region, MMSE cognitive function test score and cIMT measurements . Means and SD and sample size of cIMT and MMSE were extracted from the included papers. If mean and SD were not provided, median and range or other measures of central tendency and dispersion were extracted. Maximum cIMT was extracted if the mean value was not reported in the studies. If necessary, ORs for cIMT were converted to be per 1 mm IMT. If the percentage of people with plaques was reported, the total number was calculated. For studies in which stroke sufferers were included in the sample size, the total number of subjects were recalculated by removing subjects with stroke. The sample size was extracted for which cIMT and MMSE were measured. If different numbers of subject underwent cIMT and MMSE assessments, the lower number was recorded. The percentage of female subjects was extracted. If the total number of females was provided, the percentage of females from the overall sample size was calculated. The data were collected from baseline values or nearest value to the baseline if baseline was not available. If there were data from the article that was deficient, the authors were contacted by email. The form for eligibility of papers is available in online supplemental file 3.

## Patient and public involvement

This study does not involve human participants and did not require ethical review.

## Meta-analysis

A meta-analysis was planned using OR with 95% CI as the estimates of the strength of association between carotid atherosclerosis and MMSE. If other measures of dispersion were reported these were converted into 95% CI. ORs were chosen as recommended by Borenstein *et al*[26] as this was the effect size measure most commonly presented in the identified studies and also has the advantage of symmetry for outcome definition.[27] The two measures of carotid atherosclerosis used for meta-analysis were presence of carotid plaque and cIMT as these were the most common exposures examined in identified studies. The outcome, MMSE was categorised into normal or impaired based on a cut-point of 24, since this was the most common threshold used for analysis in the papers identified. To facilitate comparison of effect sizes from longitudinal studies with those reported by cross-sectional studies HRs or risk ratios were converted to OR using the equation described by Grant.[28] Random effects meta-analysis was performed using Stata/SE Statistical Software V.16.1 (StataCorp) using the meta suite of commands; weighting was by the inverse variance method. Q-test and

$I^2$ statistics were used to assessed heterogeneity between studies. We used the Egger's test and funnel plots to look for the possibility of publication bias.

## Quality assessment

The quality of included studies was scored using a modified seven-point criteria were derived from Newcastle-Ottawa scale.[29] Scores covered the (1) representativeness of the target sample, (2) non-response satisfactorily dealt with, (3) use of a validated measurement tool, (4) relevant confounders measured, (5) appropriate assessment of outcome, (6) appropriate statistical analysis clearly described and (7) reporting missing data if present. Each criterion was assigned one point if met. The quality assessment was performed by two researchers independently using standardised forms and any discrepancies were resolved by consensus. The Newcastle-Ottawa scale is provided in online supplemental file 4.

## RESULTS

The PRISMA flow chart illustrating the study selection process is shown in figure 1. We identified 378 potential studies from the electronic databases (Medline and Embase) after exclusion of duplicates. Title and abstract screening resulted in 48 records remaining to be reviewed for eligibility in full text. Of those studies, 17 were excluded with reasons. The reasons were: no cIMT measurement (n=4), or no MMSE test data (n=9) or use of a modified version of MMSE (n=3), review articles, conference abstract, letters to the editor or case reports (n=1).

## Characteristics studies included in the systematic review

The total number of participants in the 31 included studies was 26 178, with an age range from 35 to 95 years. Twenty of the studies were cross-sectional (online supplemental table S1A); eight were longitudinal (online supplemental table S1B). The remaining three studies included a mixture of cross-sectional and longitudinal data (online supplemental table S1C). The geographical distribution of the studies was diverse: with five from Italy, five from Japan, five from USA, three from France, two from Brazil, two from China, three from UK and one from each of Egypt, Netherlands, Norway, Serbia, Sweden and Uganda. One study failed to mention the geographical location. Only one study[30] failed to report the patients' health conditions and risk factors, such as diabetes mellitus, hypertension and chronic kidney disease. Two studies included only male participants,[31 32] and one study included only female participants,[33] the remaining studies included both male and female participants. Details of the key exposure measures are shown in online supplemental table S2. Twenty-nine studies reported the mean of cIMT measurement.[30–58] Fourteen studies did not mention the presence of carotid plaques.[31 33–36 41 45 48 49 51 52 54–56] Seven studies did not provide MMSE scores.[44 45 50 52 55 58 59]

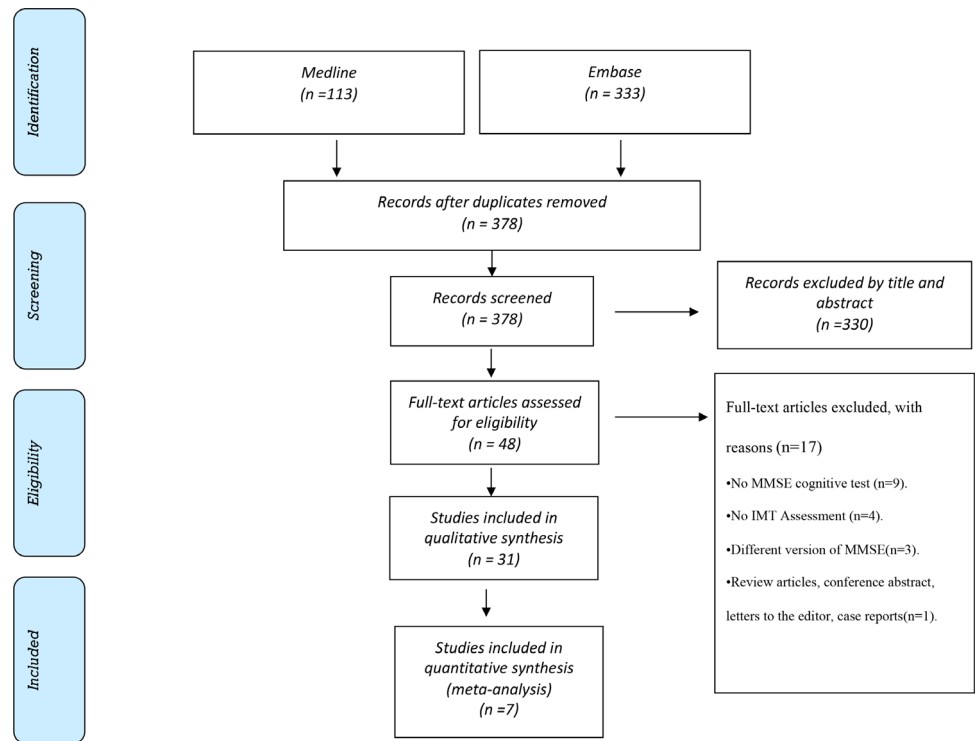

**Figure 1** PRISMA flow chart of the strategy used to select articles for review. IMT, intima–media thickness; MMSE, Mini-Mental State Examination; PRISMA, Preferred Reporting Items for Systematic Reviews and Meta-Analyses.

## Associations between atherosclerosis measures and MMSE score

### Cross-sectional associations

Presence of carotid plaques was associated with poorer cognitive function in three studies.[40 58 59] Auperin et al[30] reported an association between higher prevalence of carotid plaque and lower MMSE in men, but not in women. Another study[35] concluded that the association between cIMT and cognitive function was observed only in a low socioeconomic status group. Another study[42] found a significant correlation between higher cIMT and plaques with poor MMSE score in people with evidence of vascular cognitive decline, but not in a group of people with Alzheimer's disease, and Watanabe et al[32] reported that vascular dementia patients were more likely to have low MMSE score with thicker cIMT and frequent presences of carotid plaques. Ten cross-sectional studies[34 36 37 41 43 46 48 51 56 60] reported that cIMT was not associated with MMSE, although some of these studies reported associations with a selected aspect of cognitive performance.

### Longitudinal associations

Zhong et al reported an association between higher cIMT and reduction in MMSE score after 10 years of follow-up but not after 5 years; a similar association between presence of carotid plaque and change in MMSE was not observed at any follow-up interval.[44] Rouch et al found that cIMT and presence of carotid plaques were associated with progression to dementia in univariable analyses, but neither predicted progression after multivariable adjustment.[57] Another study found that the plaque index, and cIMT was predictive of an increased risk of conversion of MCI to dementia including after multivariable adjustment using a backwards stepwise method.[61] While Carcaillon et al reported that there was an association between presence of carotid plaque in two sites or more and incidence of dementia, they observed no association between cIMT measured at plaque-free sites with incident dementia.[50] Silvestrini et al found that baseline cIMT and increase in cIMT adjusted for baseline were associated with decline in MMSE.[39] Wendell et al found no association between baseline cIMT and change in MMSE, but observed that higher cIMT was associated with future changes in selected aspects of cognitive performance.[52] One study did not provide quantitative data about possible associations between carotid plaque or cIMT and MMSE,[54] while four other studies found no evidence that cIMT was associated with future global cognitive impairment.[33 45 47 55]

### Quality

One study received quality scores of 6 out of 7[43]; 5 received a quality score of 5 out of 7[32 47 50 57 59]; 20 a quality score of 4 out of 7[31 33–42 44–46 48 51–54 61]; 4 a quality score of 3 out of 7[30 49 56 58] and 1 a quality score of 2 out of 7.[55] Among the 31 included studies, 24 studies were adequately designed to represent the target sample. Nine publications used validated ultrasound measurement tools, 6 reported blinding assessment of outcome and blinding of outcome measurements, 29 described the statistical

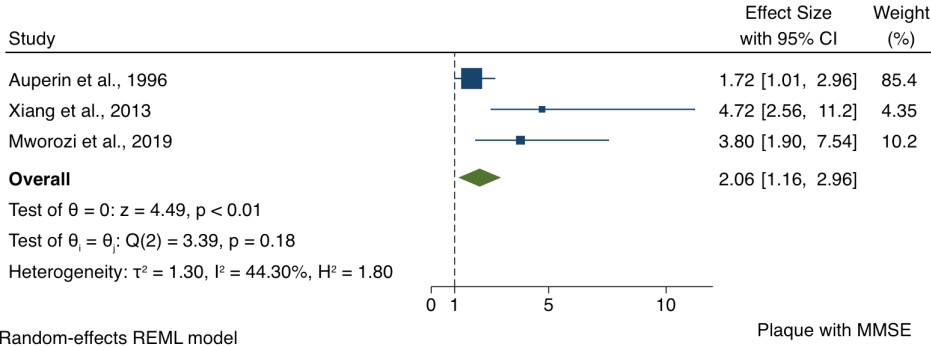

**Figure 2** Forest plots of the association between carotid plaques and Mini-Mental State Examination. IMT, intima–media thickness. REML, restricted maximum likelihood.

analysis satisfactorily. Further details are provided in online supplemental table S3.

## Meta-analysis

Seven of the 31 studies reported data that were suitable for inclusion in a meta-analysis.[30 31 41 44 46 55 58] All included studies were cross-sectional, and all but one drew from population samples. Of all studies, three reported associations between carotid plaque and impaired MMSE,[30 46 58] and six reported the association between cIMT and impaired MMSE.[30 31 41 46 55 61] There was only one suitable longitudinal study,[44] so a meta-analysis was not attempted for this category of study.

### Carotid plaque and MMSE meta-analysis

Meta-analysis showed an association between presence of carotid plaque and impaired MMSE. The overall OR for the association between carotid plaque and impaired MMSE was 2.06 (95% CI 1.16 to 2.96) for presence of plaque, and with Z-score=4.49, p<0.01 (figure 2).

### Association between cIMT and MMSE

Meta-analysis showed an association between cIMT and impaired MMSE. The overall OR for the association between cIMT and impaired MMSE was 1.13 (95% CI

1.04 to 1.22) per 1 mm cIMT, with Z-score=24.7, p<0.01 (figure 3).

### Publication bias

Egger's test showed evidence of bias in studies assessing between carotid plaque and MMSE (p=0.02), but not in cIMT with MMSE (p=0.2) (figure 4A,B).

## DISCUSSION

This systematic review examined and quantitated the evidence regarding a possible association between measures of atherosclerosis in the carotid arteries by ultrasound and generalised impairment of cognitive function assessed using the MMSE. From the limited studies available, our analysis showed there is some evidence of a cross-sectional association between carotid artery atherosclerosis and poorer MMSE, with the majority of studies finding an association. However, a sizeable proportion of studies did not find evidence for an association between cIMT and MMSE score. These discrepancies could be due to inadequate sample size, sample selection or the low sensitivity of MMSE for detection of MCI.

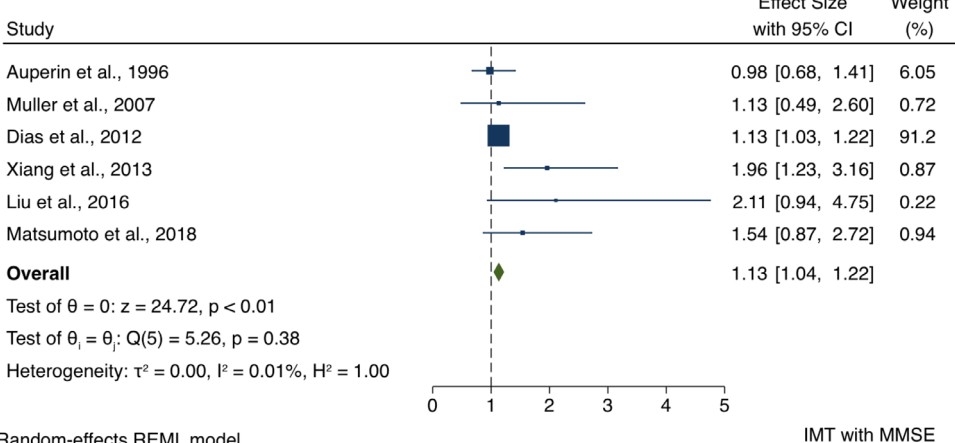

**Figure 3** Forest plots of the association between carotid intima–media thickness (IMT) and Mini-Mental State Examination (MMSE). REML, restricted maximum likelihood.

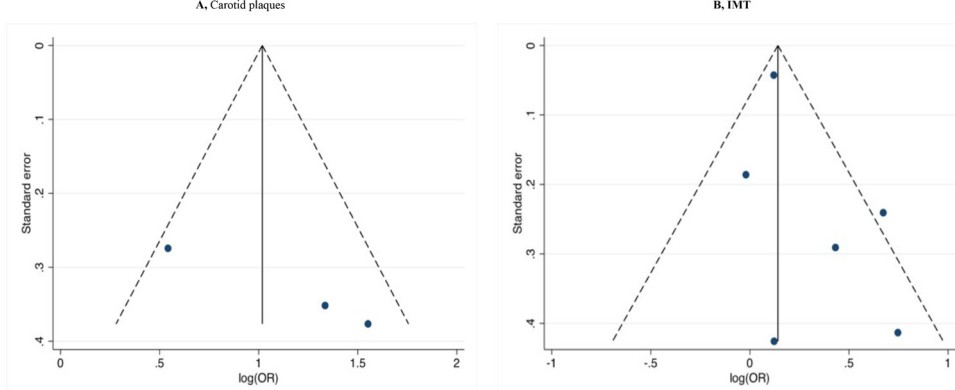

**Figure 4** Funnel plots for each outcome evaluating publication bias. (A) Studies assessed the association between carotid plaque and Mini-Mental State Examination (MMSE) shows evidence of bias (p=0.02). (B) Studies assessed the association between carotid intima–media thickness and MMSE shows no evidence of bias (p=0.2).

Risk factors for CVD are related to cognitive function[62–65] and many factors could link carotid atherosclerosis with cognition. These include thromboembolic consequences of carotid disease, small vessel disease and cerebral perfusion abnormalities, shared genetic predisposition, or shared risk factors over the life course. A recent study that observed a relationship between cIMT and MMSE, memory and executive function, also found an association between cIMT and decrease in brain matter volumes and cerebral hypoperfusion.[66]

Few studies provided information regarding factors, such as age, gender, socioeconomic position (SEP) or pre-existing CVD, that might modify any relationship between carotid atherosclerosis and cognitive impairment, so it was not possible to formally examine effect modification by meta-analysis. One study reported that there was a moderate association between presence of plaques and impaired MMSE in men, but no association in women.[30] In contrast, many studies failed to report evidence of sex differences in the association between carotid atherosclerosis and cognitive impairment,[67–69] although it was often not clear whether such an interaction was sought. Limited evidence suggests that SEP, which may include income, education and social supports may modify the association between atherosclerosis and cognitive performance. In the Whitehall II study of UK-based civil servants,[35] an association between higher cIMT and poorer cognitive function was only observed in individuals classified as low SEP. Another study done in USA,[52] reported that the association between cIMT and measures of cognition differed as a function of race and socioeconomic status.

## Limitations

Our systematic review has several limitations. Restricting the search to literature published in the English language only may have reduced the number of citations identified. The search was performed to cover publications up to 1 May 2020 and information published after that date will not have been included. The studies included in the review were drawn from diverse geographical locations, but while there was no obvious difference by location it was not possible to formally examine this question. Relatedly most studies did not report on the ethnicity of the participants, and it was therefore not possible to conclude anything about potential ethnic differences in associations. Similarly the lack of availability of individual participant data and the limitation of the summary data available restricted the analyses that could be performed. The primary aim of some of the studies reviewed was not to assess the association between atherosclerosis in carotid arteries and cognitive function. Adjustment for potential confounders was quite variable and inconsistent across studies and it is likely that this will contribute to heterogeneity between studies. There was also substantial heterogeneity with regard to atherosclerosis measurements; for example, some studies provided more information on plaque burden (eg, number, thickness or location of plaques) but this analysis was too infrequent and inconsistent across studies to allow synthesis of these data and consequently presence of plaque was chosen as the exposure of interest. There were also differences in scanning techniques between studies, and most of studies reviewed did not perform a comprehensive carotid scan. The use of B-mode ultrasound to detect carotid atherosclerosis has well-recognised limitations[10 12] and more advanced ultrasound imaging including three-dimensional ultrasound,[70] or other imaging techniques, such as MRI[71] or PET,[72] that better quantify and characterise vulnerable atherosclerosis may have advantages. Similarly, use of the MMSE has several limitations, including non-linearity, a floor effect in advanced dementia, a ceiling effect in very mild disease, and bias in people with little formal education or in non-English-speaking groups.[18–20] Also, MMSE only provides only a limited insight into the complex process of cognitive decline and use of a wider range of tools, for example assessment of a wider range of cognitive domains including executive functioning and psychomotor speed, would be worthwhile in future studies.

## CONCLUSION

This meta-analysis and systematic review provide some evidence in favour of an association between cIMT and cognitive function in cross-sectional studies. Estimates of the association between plaques and poor cognition in cross-sectional studies were too imprecise to draw firm conclusions; while evidence from studies of longitudinal associations between carotid atherosclerosis and cognitive function is limited.

**Author affiliations**
¹Diagnostic Radiology, Faculty of Applied Medical Sciences, King Abdulaziz University, Jeddah, Makkah, Saudi Arabia
²MRC Unit for Lifelong Health and Aging, UCL Institute of Cardiovascular Science, University College London, London, UK
³College of Applied Medial Sciences, Imam Abdulrahman Bin Faisal University, Dammam, Saudi Arabia
⁴Department of Public Health, College of Health Sciences, Saudi Electronic University, Riyadh, Saudi Arabia
⁵Social Research Institute, UCL Institute of Education, University College London, London, UK
⁶Department of Population Science & Experimental Medicine, UCL Institute of Cardiovascular Science, University College London, London, UK

**Correction notice** This article has been corrected since it was published Online First. One of the author's affiliation has been updated.

**Contributors** RA, AH, RH and MR developed the idea and designed the study protocol. RA, LAS and SRS designed and wrote the search strategy and data extraction. RA, MA and SRS planned the statistical analysis and results interpretation. NC, RA, RH, MR and AH provided critical insights. AH acts as guarantor for the content. All authors approved and contributed to the final written manuscript.

**Funding** RA is supported by a PhD scholarship grant from King Abdulaziz University. NC received support from the National Institute for Health Research University College London Hospitals Biomedical Research Centre and works in a unit that receives support from the UK Medical Research Council. AH receives support from the British Heart Foundation, the Economic and Social Research Council (ESRC), the Horizon 2020 Framework Programme of the European Union, the National Institute on Aging, the National Institute for Health Research University College London Hospitals Biomedical Research Centre, the UK Medical Research Council, the Wellcome Trust, and works in a unit that receives support from the UK Medical Research Council. MR is supported by UK Medical Research Council grants MC_UU_12019/1 (Enhancing NSHD) and /3 (Mental Ageing).

**Competing interests** None declared.

**Patient and public involvement** Patients and/or the public were not involved in the design, or conduct, or reporting, or dissemination plans of this research.

**Patient consent for publication** Not applicable.

**Ethics approval** This study is for conducting a systematic review and meta-analysis hence no ethical approval was required.

**Provenance and peer review** Not commissioned; externally peer reviewed.

**Data availability statement** Data are available on reasonable request.

**ORCID iDs**
Rayan Anbar http://orcid.org/0000-0002-1198-4231
Salahaden R Sultan http://orcid.org/0000-0001-9981-6138
Lamia Al Saikhan http://orcid.org/0000-0002-9951-8832
Mohammed Alkharaiji http://orcid.org/0000-0002-8135-0939
Alun Hughes http://orcid.org/0000-0001-5432-5271

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
