## [Reviewer comments · BMJ Open]

ARTICLE DETAILS

TITLE (PROVISIONAL)	Is Carotid Artery Atherosclerosis Associated with Poor Cognitive Function Assessed using the Mini-Mental State Examination? a systematic review and meta-analysis
AUTHORS	Anbar, Rayan; Sultan, Salahaden; Al Saikhan, Lamia; Alkharajji, Mohammed; Chaturvedi, Nishi; Hardy, Rebecca; Richards, M; Hughes, A

VERSION 1 – REVIEW

REVIEWER	Atsushi Iwata University of Tokyo
REVIEW RETURNED	15-Aug-2021

GENERAL COMMENTS	Anbar et al performed a systematic review and meta-analysis of papers regarding carotid atherosclerosis and MMSE. They found that the
---

REVIEWER	Abdul-Majeed Salmasi Imperial College London, cardiology
REVIEW RETURNED	05-Dec-2021

GENERAL COMMENTS	1. The word "Adult" used in the article needs to be defined.2. How was carotid artery plaque burden defined and its extent was included as a criterion for comparison? If accurate assessment of plaque burden was not referred to in the reviewed articles, this should be mentioned in the Limitations section3. Reference Auperin et al 1996 in Table 1A: there was a discrepancy between men and women in the association of carotid atherosclerosis and MMSE. How was this included in the meta-analysis?4. Diverse geographical regions of the studies: Was there a difference in the association between carotid atherosclerosis and MMSE in different regions? Did the author notice racial diversity in the studies? This must be addressed5. There was a wide age range 35-95 of the study population. Was there a preference in the association between carotid atherosclerosis and MMSE with age?
---

REVIEWER	Simona Lattanzi Universita Politecnica delle Marche
REVIEW RETURNED	24-Dec-2021

GENERAL COMMENTS	This was a systematic review with meta-analysis aimed to evaluate the associations between carotid atherosclerosis and global cognitive function as assessed by the mini-mental state examination. The study is interesting, addresses a very important topic and is
--

	overall well-organized. There are, however, some issues to address further. It would be important to briefly highlight the pathophysiology of cognitive impairment in patients with carotid atherosclerosis, and in particular the evidence that cerebral hemodynamics may be an independent and potentially reversible determinant of cognitive dysfunction (Ref. Neurocognitive functioning and cerebrovascular reactivity after carotid endarterectomy. Neurology 2018; Predictors of cognitive functioning after carotid revascularization. J Neurol Sci 2019).
--	---

REVIEWER	Brynjar Fure Örebro universitet
REVIEW RETURNED	30-Jan-2022

GENERAL COMMENTS	Thank you for the opportunity to review this systematic review and meta-analysis by Anbar et al. The topic of this manuscript is of interest to clinicians and researchers in general practice, internal medicine, neurology, geriatrics, psychiatry and other specialties. My inputs to the manuscript: Introduction The introduction includes a presentation of atherosclerosis and cognitive decline. In line 35/36 of the introduction, the authors state that: "Mild Cognitive Impairment (MCI), dementia and Alzheimer's are the most common types of cognitive dysfunction". This statement is somewhat confusing since MCI and dementia are main groups of cognitive impairment (in DSM-V based on clinical criteria such as dependence in ADL) whereas Alzheimer's disease is an etiological subtype of cognitive impairment that can cause both dementia and MCI. I suggest this sentence is rewritten. The paragraph on cognitive impairment introduces the concepts dementia and MCI without presenting definitions of these condition. It would be a strength if dementia and MCI were defined. Methods Search strategy The search is nearly 2 years old – this should be mentioned as a weakness, since new studies might have been published after 01/05/2020. Was an information specialist/librarian involved in preparing or conducting the literature search? Exclusion criteria Exclusion criterion 3 can be removed since animals were not mentioned in the inclusion; inclusion criterion one states that studies should include (1) Adults, i.e. humans. Meta-analysis The authors explain that hazard ratios and risk ratios were converted to odds ratios according to the equation described by Grant. This might be controversial, as some statisticians and authors of meta-analyses regard such conversions as questionable. In addition, as far as I understand it, Grant describes conversion from odds ratios to risk ratios, and not vice versa. If the authors regard conversions from hazard ratios and risk ratios to odds ratios as statistically acceptable, it would be a strength if they could refer to other publications where this method has been used. In this connection, it may also be a challenge that the risk ratios and hazard ratios that were converted
---

	to odds ratios probably were adjusted for different potential confounders in different studies – thereby introducing more heterogeneity in the analyses. It is not quite clear to me how the meta-analyses were performed. Did the authors introduce logarithmic values of the odds ratios into the analyses? As Figures 2 and 3 contain no information on this, it would be useful if the authors explain more in detail which values of odds ratios that were used in the analyses. How were the weights calculated in the meta-analyses? I am a bit surprised that some of the studies seem to have no impact on the overall result with weights as low as 0.22 (Liu et al). It would be useful if a table reporting results of the quality evaluation using checklists was attached as a supplementary. Results Table 1A and 1C are, in my opinion, nicely presented and easy to read. Table 2 contains information on the exposure measures; I suggest removing the information on which ultrasound company that had produced the machines used in different studies, as this probably is of very limited interest to most readers. Figure 2 and 3 present results as odds ratios, as I understand it. This should be explained more clearly in the Figures. Discussion The authors discuss the weaknesses of the MMSE in the Introduction. Would it be more logical to move this discussion to the discussion section? One major weakness of MMSE, that is not mentioned by the authors, is that the MMSE does not evaluate the domains executive functioning and psychomotor speed. Many researchers and clinicians regard impairments within these domains as the most important cognitive deficits in vascular cognitive impairments. It would be useful if this weakness was mentioned. In paragraph three of the discussion section (lines 44/45), the abbreviation SES is used. Is this a typo for socioeconomic position (SEP)?
--	--

VERSION 1 – AUTHOR RESPONSE

Reviewer: 2

Comments to the Author:

1. The word “Adult” used in the article needs to be defined.

We have defined adult according to the World Health Organization (WHO), i.e. a person older than 19 years of age (Canêo LF, Neirotti R. The Importance of the Proper Definition of Adulthood: What is and What is Not Included in a Scientific Publication. Braz J Cardiovasc Surg 2017;32(1):60-60. doi: 10.21470/1678-9741-2016-0049) (page 6 line 11).

2. How was carotid artery plaque burden defined and its extent was included as a criterion for comparison? If accurate assessment of plaque burden was not referred to in the reviewed articles, this should be mentioned in the Limitations section.

Carotid plaques were assessed in a number of ways in the identified studies, by far the most common being presence (or absence) of plaque. In some studies plaque burden was assessed in more quantitative ways (e.g. number or location of plaques) but this analysis was too infrequent and inconsistent across studies to allow synthesis of these data and consequently presence of plaque was

chosen as the exposure of interest. We have now clarified this in methods (page 7 lines 20-22) and added a comment regarding this issue in discussion (page 12 lines 12-16)

3. Reference Auperin et al 1996 in Table 1A: there was a discrepancy between men and women in the association of carotid atherosclerosis and MMSE. How was this included in the meta-analysis? It was not possible to include data from Auperin et al. stratified by sex in this analysis, only data pooled across sexes. In the main, this was because other studies did not provide data stratified by sex. However, we also noted that no test of statistical interaction was performed by Auperin et al. (they only observed that associations were statistically significant in men and not in women) and hence it is not clear whether there was strong evidence for a difference by sex in this study or whether stratification by sex had been pre-specified in the analysis plan. We have added a brief comment on page 11 line 11- 14 in response to the reviewer's comment.

4. Diverse geographical regions of the studies: Was there a difference in the association between carotid atherosclerosis and MMSE in different regions? Did the author notice racial diversity in the studies? This must be addressed

There was no obvious difference between different geographical regions although given the diversity and the limited number of studies available for meta-analysis it was not possible to formally examine this question. Relatedly most studies did not report on the ethnicity of the participants, and it was therefore not possible to conclude anything about potential ethnic differences in associations. We have added these issues as limitations of the study on page 12 line 3 – 6.

5. There was a wide age range 35-95 of the study population. Was there a preference in the association between carotid atherosclerosis and MMSE with age?

While the age range of participants in the studies was wide, most studies predominantly included older people such that their average age did not generally differ substantially. This is not unexpected given that both atherosclerosis and cognitive impairment are more likely with increasing age. The question of whether age modified the relationship between atherosclerosis and cognitive performance is interesting, but it was not possible to address this question using the data available. Of the 5 cross-sectional studies available for meta-analysis, one did not report the average age, while three only reported the average age of subgroups analysed and only one presented the overall average age of the sample. With only four samples available (and given the limitations in age reported) it was not considered appropriate to undertake a meta-regression. We have briefly commented on this (and the difficulty in performing any stratified analyses) on page 11-12 lines 32 .

Reviewer: 3

Comments to the Author:

This was a systematic review with meta-analysis aimed to evaluate the associations between carotid atherosclerosis and global cognitive function as assessed by the mini-mental state examination. The study is interesting, addresses a very important topic and is overall well-organized. There are, however, some issues to address further.

It would be important to briefly highlight the pathophysiology of cognitive impairment in patients with carotid atherosclerosis, and in particular the evidence that cerebral hemodynamics may be an independent and potentially reversible determinant of cognitive dysfunction (Ref. Neurocognitive functioning and cerebrovascular reactivity after carotid endarterectomy. *Neurology* 2018; Predictors of cognitive functioning after carotid revascularization. *J Neurol Sci* 2019).

Thank you for this suggestion, this point and the recommended citation has been added to the revised manuscript (page 4 lines 11-13).

Reviewer: 4

Comments to the Author:

Thank you for the opportunity to review this systematic review and meta-analysis by Anbar et al. The topic of this manuscript is of interest to clinicians and researchers in general practice, internal medicine, neurology, geriatrics, psychiatry and other specialties.

My inputs to the manuscript:

The introduction includes a presentation of atherosclerosis and cognitive decline. In line 35/36 of the introduction, the authors state that: "Mild Cognitive Impairment (MCI), dementia and Alzheimer's are the most common types of cognitive dysfunction". This statement is somewhat confusing since MCI and dementia are main groups of cognitive impairment (in DSM-V based on clinical criteria such as dependence in ADL) whereas Alzheimer's disease is an etiological subtype of cognitive impairment that can cause both dementia and MCI. I suggest this sentence is rewritten. The paragraph on cognitive impairment introduces the concepts dementia and MCI without presenting definitions of these condition. It would be a strength if dementia and MCI were defined.

Thank you. We apologise - this sentence was indeed confusing, and inclusion of definitions of MCI and dementia is a very helpful suggestion. We have extensively re-written this paragraph to address these two points.

Methods

Search strategy

The search is nearly 2 years old – this should be mentioned as a weakness, since new studies might have been published after 01/05/2020.

This is a fair point and we have added it to the limitations on page 11 lines 25-26 (just for the reviewer's information we should point out that this paper was initially submitted to the BMJ on 04 Jul 2021 but the allocation of reviewers to the review process was extremely delayed, presumably due to COVID).

Was an information specialist/librarian involved in preparing or conducting the literature search?

Yes, advice was obtained from the information specialist at UCL library, including training and advice about creating a research strategy. We have added a comment to this effect in the revised manuscript (page 6 line 4).

Exclusion criteria

Exclusion criterion 3 can be removed since animals were not mentioned in the inclusion; inclusion criterion one states that studies should include (1) Adults, i.e. humans.

Thank you, this has been done (page 6 line 16-22 and line 11-12)

Meta-analysis

The authors explain that hazard ratios and risk ratios were converted to odds ratios according to the equation described by Grant. This might be controversial, as some statisticians and authors of meta-analyses regard such conversions as questionable. In addition, as far as I understand it, Grant describes conversion from odds ratios to risk ratios, and not vice versa. If the authors regard conversions from hazard ratios and risk ratios to odds ratios as statistically acceptable, it would be a strength if they could refer to other publications where this method has been used. In this connection, it may also be a challenge that the risk ratios and hazard ratios that were converted to odds ratios probably were adjusted for different potential confounders in different studies – thereby introducing more heterogeneity in the analyses.

Thank you. We accept that the choice of common effect size for meta-analysis is somewhat debatable. We followed the recommendation of Borenstein et al. and based the choice of odds ratio based on their being the most common effect size reported in identified papers. They also have the additional advantage of symmetry, although this was not decisive in our choice. Grant's paper does describe the conversion from odds ratios to risk ratios although the reverse transformation is implicit in the equation. We agree that inconsistent adjustment across studies is a problem (we have now added this issue to limitations on page lines). To some extent this is a problem irrespective of what effect size is used in that it introduces variable adjustment for confounding. However, the issue of non-collapsibility of odds ratios should not affect risk or hazard ratios and might be considered a disadvantage of this choice.

It is not quite clear to me how the meta-analyses were performed. Did the authors introduce logarithmic values of the odds ratios into the analyses? As Figures 2 and 3 contain no information on this, it would be useful if the authors explain more in detail which values of odds ratios that were used in the analyses.

We have added a little more detail on this, namely that we used the meta suite of programs in Stata 16. This analysis method is based on the log odds ratios but we presented the odds ratio for ease of interpretation.

How were the weights calculated in the meta-analyses? I am a bit surprised that some of the studies seem to have no impact on the overall result with weights as low as 0.22 (Liu et al).

Weights were assigned using the inverse-variance estimation method, which is the default in meta in Stata. We have added this information to the statistical methods (page 7 lines 26-7).

It would be useful if a table reporting results of the quality evaluation using checklists was attached as a supplementary.

These data have been added to the revised manuscript as supplementary table3

Results

Table 1A and 1C are, in my opinion, nicely presented and easy to read.

Table 2 contains information on the exposure measures; I suggest removing the information on which ultrasound company that had produced the machines used in different studies, as this probably is of very limited interest to most readers.

We have adopted this suggestion.

Figure 2 and 3 present results as odds ratios, as I understand it. This should be explained more clearly in the Figures.

This has been clarified as requested.

Discussion

The authors discuss the weaknesses of the MMSE in the Introduction. Would it be more logical to move this discussion to the discussion section?

We have moved this to the discussion as suggested (page 12 lines 15-18).

One major weakness of MMSE, that is not mentioned by the authors, is that the MMSE does not evaluate the domains executive functioning and psychomotor speed. Many researchers and clinicians regard impairments within these domains as the most important cognitive deficits in vascular cognitive impairments. It would be useful if this weakness was mentioned.

We have added this important point to the discussion.

In paragraph three of the discussion section (lines 44/45), the abbreviation SES is used. Is this a typo for socioeconomic position (SEP)?

We have corrected this typo.

VERSION 2 – REVIEW

REVIEWER	Brynjar Fure Örebro universitet
REVIEW RETURNED	26-Mar-2022
GENERAL COMMENTS	Thank you for the possibility to read a revised version of this systemic review and meta-analysis. In my opinion, the authors have addressed all inputs in a satisfactory way. I have no further inputs or concerns regarding this manuscript. The present manuscript presents valuable knowledge on the relationship between vascular disease and cognitive impairments.